# *"Just a knife wound this week, nothing too painful"*: An ethnographic exploration of how primary care patients experiencing homelessness view their own health and healthcare

**Carolyn Ingram**[1]*, **Conor Buggy**[1], **Isobel MacNamara**[1,2], **Carla Perrotta**[1]

1 Public Health, School of Public Health, Physiotherapy, and Sports Science, University College Dublin, Dublin, Ireland, 2 School of Medicine, University College Dublin, Dublin, Ireland

* Carolyn.ingram@ucd.ie

**Data Availability Statement:** All relevant data are within the manuscript and its Supporting information files.

## Abstract

Community health needs assessments (CHNA) involving qualitative techniques help tailor health services to the specific needs of the population groups for whom they are designed. In light of increasing health disparities amongst people experiencing homelessness (PEH)—and to ensure the integration of their voices into a larger CHNA—this study used an ethnographic approach grounded in a social constructivist research paradigm to explore the perspectives of PEH attending a primary care and addiction service in Ireland on their priority health and healthcare needs. Participant observations and informal interviews were conducted with clients experiencing homelessness attending the service for three hours every Monday morning between October 2022 and April 2023. Field note data from active participant observation and informal conversations were collected, anonymised, and analysed using inductive thematic analysis in accordance with the Declaration of Helsinki and the researchers' institutional Research Ethics Committee. Three main themes emerged from the analysis: self-identified priorities, satisfaction with health services, and migrant health. Clients' priority concerns relate to their mental health and personal safety, strengthening ties with children and families, finding a sense of purpose, and feeling better physically. These challenges differ from those of the general population in terms of their severity observed both prior to and during experiences of homelessness, coupled with disproportionately high levels of loss, fear, pain, fatigue, social stigma and other barriers to accessing satisfactory housing. In terms of services, clients are satisfied with their ability to access primary care and harm reduction in a social environment where positive exchanges with friends and providers take place. Conversely, barriers to accessing mental health and addiction services persist including the internalised belief that one is beyond help, lack of access to information on available services and their entry requirements, and lingering stigma within a health system that treats addiction as separate to health. Moving forward, health practitioners may consider holding more regular and open conversations with clients experiencing homelessness about the care they are receiving, its rationale, and whether or not changes

**Funding:** The author(s) received no specific funding for this work.

**Competing interests:** The authors have declared that no competing interests exist.

are desired that can be safely made. The health needs of migrants and asylum seekers entering homelessness in Ireland are urgent and should be prioritised in future research.

## Introduction

Rising homelessness levels across Europe have brought increased attention to the health consequences of precarious housing. People experiencing homelessness (PEH) are at higher risk of premature mortality, disability, and chronic conditions but face reduced access to health services [1, 2]. A health equity perspective—the belief that everyone has the right to a fair and just opportunity to attain their full potential for health—emphasises that housing-related health disparities are unnecessary and avoidable. Rather than the result of biology, they result from social and economic processes that create / recreate differences in access to housing, health services, and in turn health [3]. Medical and public health practitioners acknowledge that advancing health equity requires not only addressing the downstream drivers of homelessness, but also improving equitable access to healthcare for PEH who experience myriad barriers to care due to complicated application processes or appointment systems, long queues, judgement from healthcare providers, competing priorities (e.g., finding food or a place to sleep), and internal barriers like fear, embarrassment, hopelessness, and poor self-esteem [4].

Ireland is unique in Europe in the severity of spikes in homelessness over the last decade [5]. Across the Dublin region (population 1.4 million) where most national homelessness is concentrated, approximately 7,000 adults and 3,000 children are currently residing in emergency accommodation [6]. An estimated 118 individuals are sleeping rough [7], and at least 6% of the city's residents are experiencing hidden homelessness (i.e., sleeping cars, in squats, on the floors or sofas of family and friends, or in unsafe accommodation) [8]. Drug and alcohol use is the primary driver of homelessness in Ireland and remains a leading cause of death amongst PEH [9]. In response, and to circumvent known barriers to care, a network of multidisciplinary primary health care and addiction services provide specialised services for PEH across Dublin in partnership with the Health Services Executive Ireland (HSE) [10]. These services have drastically improved access to primary care and harm reduction in recent years and Ireland is seen to be a leader in Europe in its provision of innovative healthcare at the community level [10, 11].

Despite these improvements, in autumn 2022, multiple factors highlighted a need to identify priority health concerns amongst PEH who are accessing homeless health services in Ireland. First, despite increased access to primary care, evidence shows that disparities in mortality between homeless and housed populations continue to worsen [12]. Second, high rates of Covid-19 vaccine hesitancy amongst marginalised groups stressed the extent to which public health campaigns that fail to identify the priority needs and concerns of those whom they affect are counterproductive, especially amongst communities used to being let down by health and other government services [13]. Finally, the Covid-19 pandemic acted as a catalyst for change in the accessibility of harm reduction measures for PEH who use drugs. Positive changes—namely increased access to methadone and naloxone—demonstrated the capacity for policymakers to remove barriers to accessing care in response to a strong public health argument [14]. Researchers thus saw an opportunity to provide further evidence on how and where to allocate health and social service resources effectively.

Community health needs assessments (CHNA) that include qualitative techniques are designed to help tailor health services to the needs of specific population groups [15]. The

integration of voices from community members into these assessments enriches the CHNA process in several ways [16]. A social constructivist paradigm assumes the world is made up of multiple realities, local truths, and subjective experiences [17]. From this lens, interacting with community members provides otherwise unobservable data that may help bridge the divide between theory and practice [18]. Qualitative needs assessment approaches involving community members can help improve quality by identifying gaps or current problems in care and help focus initiatives on what is important to those for whom they are designed [16]. Labonte and Robinson [17] describe what may follow should health researchers and practitioners overlook the needs most important to a community, *"If we fail to start with what is close to people's hearts by imposing our notions of health concerns over theirs, we risk several disabling effects. We may be irrelevant to the lives and conditions of many persons . . . We may further their experience of powerlessness by failing to listen and act upon concerns in their lives as they experience and name them, communicating to them that they are wrong and we are right."* To date, there remains a gap in the evidence in terms of how PEH view their own health and healthcare in Ireland [11].

To ensure the integration of community voices into a larger, qualitative CHNA [11, 16], this study used an ethnographic approach to explore the perspectives of PEH attending an urban primary care and addiction service on their priority health and healthcare needs.

## Materials and methods

### Study design

The lead researcher engaged in ethnographic fieldwork at a drop-in primary care and addiction services clinic serving primarily PEH. She initially approached the clinic founder to determine mutual interest in a research collaboration and establish a community partnership. The central question explored in this article (i.e., *What are PEH's own views on their health and healthcare needs*?) and associated research methods were then developed by the research team and clinic staff. Ethnography was the chosen methodology grounded in a social constructivist research paradigm. Ethnographic methods allowed for a focus on holism [19]. Rather than predefining and restricting the areas the researcher considered relevant to the health of PEH, a broader approach allowed her to identify and understand client concerns she could not have expected. A social constructivist paradigm holds that multiple realities exist, each an intangible construction rooted in people's experiences with everyday life, how they remember them, and make sense of them [20]. From this lens, the researcher could become a research instrument for generating a "consensus construction of reality" more informed and sophisticated than individual constructions [17, 20].

The ethnographic approach to fieldwork combined active participant observation and informal interviews—both casual and more in-depth—with clinic attendees. Participant observation was used to gain insight into naturally occurring data and events happening in the study setting and to identify how clients are accessing / using primary healthcare and addiction services. The researcher adopted the role of the 'active participant observer'; her research was explicit rather than covert. This allowed her to become trusted by those attending and working in the clinic over time and keep returning patients apprised of her ongoing findings and activities [21]. Observations were enriched by informal conversations that helped the researcher strengthen trust, establish a rapport, and see situations from clients' perspectives in a natural and inartificial way. Researchers argue that informal interviews build a less threatening environment, create an ease of communication, and allow for the engagement of vulnerable community members who may be less likely to participate in formal interviews or surveys [22, 23].

## Participants and setting

The target population for the ethnographic research was PEH linked with primary healthcare and/or addiction services in Ireland. The sampled population was people experiencing (or who had experienced) homelessness attending the partner drop-in primary care and addiction services clinic at least one Monday between October 2022 and April 2023. For the purposes of this study, homelessness was defined as (1) having no accommodation available that can reasonably be stayed in or living in a hospital, county home, night shelter or other such institution, and (2) being unable to provide accommodation from one's own resources [24]. Located in a densely populated urban area, the study site provides a walk-in general practitioner (GP) service on weekday mornings for patients who are experiencing homelessness or who are undocumented migrants. The clinic, funded by the charity sector and HSE, is part of a network of specialised, low-threshold homeless health services operating across Dublin, Galway, Cork, and Limerick. Key working services and needle exchange are provided on site, as are opiate substitution and—under a strict protocol—benzodiazepine detoxification and community alcohol detoxification. Patients who have a medical card (formal access to free health services through the HSE) are encouraged to seek care with their own GP.

Prior to commencing the formal observation period, the researcher visited the clinic to meet staff and familiarise herself with the setting. Clinic attendees were selected for informal interviews during the researcher's subsequent visits using purposive, critical case sampling [25]. Clinic medical or operational staff identified and introduced the researcher to potential 'critical cases' who were present in the waiting room and met the following criteria: (1) had experienced/were experiencing homelessness and potential co-occurring addiction, (2) had an established rapport with centre staff and were deemed to be of sound mental capacity, (3) expressed a voluntariness to chat with the researcher without coercion, and (4) spoke English. As the project progressed, clinic staff helped the researcher identify critical cases with yet-to-be-explored characteristics / experiences (e.g., clients of a certain age or gender, clients sleeping rough, clients in early recovery).

## Data collection

Participant observations and informal interviews were conducted for three hours every Monday morning between October 2022 and April 2023. The day was selected based on clinic staff's preferences; Mondays were deemed the least chaotic with more space for the researcher to chat with patients. Data collection took place in the form of in-depth and casual conversations, and observations. The researcher chose not to record conversations during her visits as it could create concerns about confidentiality and privacy [26]. Instead, she jotted down raw records in a notebook (during observations and in-depth conversations or after casual conversations), a technique recommended for maintaining informality and building trust with vulnerable participants [27]. To increase fidelity, the researcher reflected back perceived critical points to the participant during interviews and wrote up formal fieldnotes immediately after each visit using a four-section template to standardise data collection modelled after Gibson 2013 [28]. In the first section—Description of Activity—she recorded descriptive, open-ended notes on conversations held and observations made in the order of their occurrence. Conversations were delineated from observations using headings with the conversation type and participant's name (later pseudonym). The latter three sections–Reflections, Emerging Questions/ Analyses, Future Action—provided space for analytic notes where, distinct from the descriptive data, the researcher began to interpret what was heard and observed. When a point of uncertainty emerged during the writing up of interview data, it was marked as 'needs follow-up' and added to future actions. If the researcher crossed paths again in the clinic with the

interviewee, she asked them to clarify the point in question or used it to prompt discussion during conversations with other participants.

**In-depth conversations.**   After an introduction to a potential participant by a clinic staff member, the researcher presented herself as a PhD scholar interested in the participant's concerns and needs regarding their healthcare. If the client agreed, they were invited to accompany the researcher into an open consultation room. Informed consent for informal interviews was obtained verbally; a recommended approach for non-recorded research with people who are/have been homeless and experience significant state regulation and stigma leading to mistrust of activities that require their signature [29]. Participants were informed in more detail that the research was looking at clients' own concerns and needs regarding their healthcare, that any notes taken during and after the conversation would be fully anonymised, and that results would be submitted for scientific publication and shown to policymakers. Once informed verbal consent was obtained, the researcher led with a general question: *"Do you mind telling me a bit about how are you feeling today?"* If healthcare didn't come up naturally in conversation, the researcher would ask, when appropriate, *"and what about your healthcare, do you feel like you could be better supported in any way?"* In-depth conversations lasted between 20 minutes to 1 hour.

**Casual conversations.**   In some instances, the researcher would initiate a non-directive [30] conversation in the waiting room with the individual seated next to her, or vice versa. She learned simple techniques for determining an individual's interest in conversing; things like saying 'bless you' when someone sneezed or offering to watch someone's seat while they went to the toilet. If the person was quiet, the researcher would not try to engage them any further. If the person was interested in chatting, the researcher would follow the organic flow of conversation, documenting fully anonymised exchanges in her field notes after each visit. When appropriate, the researcher would formally introduce herself. Because she was not able to assess whether the individual was in a position to grasp and reflect on information about the study, she would not invite someone to participate in an in-depth conversation without a formal introduction from clinic staff. These conversations—by allowing the researcher to talk casually with people and hear what was on their mind—added to the richness of participant observations.

It was not uncommon for the same clients to return frequently to the clinic, either for methadone/other prescriptions or because they lived in the attached hostel. When the researcher saw an individual she had already spoken with, she made a point of saying hello and asking how that person was getting on. Any new information was recorded after the visit in her field notes. Over the course of her observations there were six individuals who became familiar enough with the researcher to call her by name and sit next to her while they waited to see a doctor.

**Observations.**   When not in conversation the researcher noted what was happening around her including client appearance, verbal behaviours and interactions, physical behaviours, and human traffic [31]. Because clients chatted openly amongst themselves or on the phone while waiting, it was possible to observe and record information on housing status, and nationality. Information on substance use, when not overheard, was observed as clients handed over urine samples, picked up methadone scripts, were visibly intoxicated, or experienced withdrawal while waiting.

## Ethical considerations

All participants were 18 years of age or older and have been given a pseudonym. Methods were performed in accordance with the guidelines and regulations outlined in the Declaration

of Helsinki and the study passed full ethical review by the University College Dublin Human Research Ethics Committee—[Sciences (HREC-LS)]. Research Ethics Reference Number (REERN): HREC LS-22-42-Ingram-Perrotta. According to recommended guidelines[32, 33], informed oral consent was obtained from all participants who met inclusion criteria and participated in non-recorded, in-depth interviews. In some instances, informed consent could not be obtained during casual conversations or participant observations due to the complex dynamics of the clinic space. In these instances, only non-identifying public data were recorded (e.g., statements made loud enough for the entire waiting room to hear) [32] and any collected data analysed and reported at the general level. Explicit consent was sought and obtained from all participants who are quoted in the manuscript. To ensure that she did not become emotionally or psychologically distressed when researching sensitive issues, the researcher regularly debriefed with community partners and research colleagues and took breaks between interviews as needed.

### Rigour and trustworthiness

Lincoln and Guba (1985)'s criteria for enhancing rigour based on credibility, transferability, dependability and confirmability were followed [34]. To ensure credibility, the lead researcher, who is formally trained in qualitative research techniques, spent a prolonged period at the study site (60 hours) and created the study protocol in collaboration with clinic partners with whom she also held regular debriefing sessions over the course of the project. The inclusion of in-depth conversations whereby purposively chosen participants were invited to speak freely about what was on their mind supported the construction of data representative of the authentic views of PEH. Choosing not to audio record participants was, by design, intended to prevent inhibitions in open discussion by ensuring that participants felt neither controlled nor examined [22]. To increase the fidelity of her notes, the researcher reflected perceived critical points back to participants during and after interviews and wrote up field notes immediately after each visit using a standardised template. Readers should nevertheless keep in mind that the data presented come from the lead researcher's own notes and memory and are subject to some degree of recall bias [23]. To ensure dependability, in addition to the description of methods presented here, the study protocol was pre-registered on Open Science Framework [35]. As part of a larger CHNA, confirmability was enhanced by triangulating study findings with previous literature published on similar cohorts in Ireland [2], and the perspectives of health and addiction professionals working with the same population during the study period [11]. The researcher's field notes template included a section for personal reflections to encourage reflexivity. Finally, detailed field note excerpts are included in S1 Dataset in addition to illustrative quotes embedded within the results to ensure that researchers can make judgements on the transferability of findings to other study populations [34].

### Analysis

In line with the social constructivist process of iteration, analysis, critique, reanalysis, and synthesis [17], data analysis took place alongside data collection. Before each clinic visit, the researcher reviewed notes written in the 'Emerging Questions/Analyses' portion of her field-notes to determine where to guide conversations should clients seek to be prompted. This allowed for the progressive focusing of interviews and observations on issues that seemed particularly important or controversial to clients. Because the development of broad analytic categories that captured participants' priority concerns and the mechanisms driving them emerged before data collection was complete, the researcher was able to run findings by other participants as a means of strengthening credibility. The researcher also regularly presented

clients' views to the wider research team and clinic partners to gain clarity on whether expressed concerns matched the realities of the services provided. Oftentimes they did, as demonstrated by this excerpt from the researcher's analytic field notes:

> *After speaking with C today, I am left in disbelief that she was expelled from the women's shelter after her partner found out where she was staying but that she was given no other option of another shelter or somewhere safe to go. Is this really the way things are handled in practice? Or did perhaps she perceive being unwelcome there or leave without mentioning her departure to staff? [Later entry] I ran this by GP 2 this morning in one of the clinic's quiet moments. He says it's not unlikely that this really happened. Domestic violence is almost expected amongst homeless women and some shelters don't accept or keep female homeless residents for fear that they are putting on an act to get a place to stay. He confirms that as limited as refuge spaces are across the country, they are even harder to come by if you are homeless and especially if drug use is involved.*

Other times, barriers were perceived by clients which did not actually exist and the researcher was able to note this distinction in her analytic field notes. The process served as an exercise in reflexivity whereby the researcher continually considered and documented fluctuating tendencies to take clients at their word or, alternatively, to enter field visits with a greater degree of scepticism after speaking with clinic staff.

Once data collection was complete, field note data (80 pages, 40,000 words) were coded and analysed according to recommended stages of trustworthy, inductive thematic analysis [36–38]. Two researchers (CI,IM) familiarised themselves with the data by re-reading and annotating the descriptive sections of the field notes. The researchers came together to discuss their initial analytical notes, thoughts, and impressions. By design, CI waited to share her sense of emerging broad analytic categories until her colleague had formed her own initial impressions of the data. The same researchers then independently coded ten pages of descriptive field notes line by line to capture the data's keywords and core messages. In line with the social constructivist paradigm, codes were not limited to traditional health or health care concepts. During this 'open coding' process, the two researchers started to identify potential themes and subthemes relating to clients' views on their health or healthcare. Descriptive field notes included both observational and informal interview data delineated by data type. When results were compared, the researchers identified consistency of codes across the data types and chose to construct one initial coding framework based on the entirety of the dataset. CI and IM coded ten more pages each before further discussing and refining the work. During this stage of peer debriefing, the researchers recognised and agreed upon three thematically distinct constructs: self-identified priorities, satisfaction with health services, and migrant health. Data from a final theme—barriers and facilitators to addiction recovery—were deemed rich enough to be reported in a separate paper. The working thematic framework was systematically applied to all transcripts by CI using Nvivo software V.11 and a final version with themes, sub-themes, codes, and illustrative quotes was shown to and agreed upon by the wider research team.

At this stage, the researchers entered into the interpretation phase to determine how the themes related to the initial research question of how PEH attending the primary care and addiction clinic understand their own health and healthcare needs. The research team met to discuss connections within and between themes and sub-themes, after which CI developed a data-driven conceptual model depicting (1) self-identified priority needs, (2) barriers to meeting those needs, and (3) proposed solutions for overcoming those barriers. Once this model was approved by the wider team, in a final step, community partners were shown a copy of the

complete coding framework and proposed conceptual model and invited to provide feedback on the accuracy of the researchers' thematic analysis and conceptualisation.

## Results

### Participant characteristics

Between October 2022 and April 2023, the researcher held in-depth conversations (N = 23) and casual conversations (N = 15) with 38 clinic attendees and recorded observations for 36 more. Participant characteristics are summarised in Table 1. Conversations and observations were distributed evenly across genders (51% men, 49% women). Most participants were between 20 and 50 years old (N = 62, 84%), with only five individuals aged 60 and older. Participants resided in homeless hostels (N = 41, 55%) or with family (N = 6, 8%). A minority of

**Table 1. Participant characteristics of 74 clients attending a drop-in primary care and addiction service clinic between October 2022 and April 2023.**

| | In-Depth Conversation | Casual Conversation* | Observation** | Total N (%) |
|---|---|---|---|---|
| Total | 23 | 15 | 36 | 74 (100) |
| **Gender** | | | | |
| Man | 11 | 8 | 19 | 38 (51) |
| Woman | 12 | 7 | 17 | 36 (49) |
| **Age Estimate** | | | | |
| 20–30 | 7 | 7 | 7 | 21 (28) |
| 30–40 | 6 | 1 | 16 | 23 (31) |
| 40–50 | 7 | 4 | 7 | 18 (24) |
| 50–60 | 1 | 2 | 4 | 7 (10) |
| 60–70 | 2 | 1 | 2 | 5 (7) |
| **Housing Status** | | | | |
| Homeless Hostel | 7 | 8 | 26 | 41 (55) |
| Sleeping Rough | 5 | 0 | 2 | 7 (10) |
| Living with Family | 5 | 0 | 1 | 6 (8) |
| Long Term Housing | 4 | 4 | 0 | 8 (11) |
| Direct Provision | 1 | 0 | 0 | 1 (1) |
| NA | 1 | 1 | 9 | 11 (15) |
| **State of Addiction** | | | | |
| No addiction | 4 | 4 | 1 | 9 (12) |
| Active addiction[1] | 14 | 21 | 17 | 52 (70)[2] |
| In recovery[3] | 5 | 2 | 2 | 9 (12)[4] |
| NA | 0 | 0 | 4 | 4 (5) |
| **Migrant Status** | | | | |
| Yes | 4 | 2 | 1 | 7 (9) |
| Irish Citizen | 19 | 13 | 35 | 67 (91) |

*Casual conversations: Participant fully controls the disclosure of information; non-public data are not reported as informed verbal consent not obtained.

**Observations: Information gathered through visual and auditory observations; non-public data are not reported as informed verbal consent not obtained.

[1] Active addiction defined as compulsive substance seeking and use [40]

[2] Approximately 32/52 (62%) patients in active addiction were observed on opioid substitution therapy (OST); approximately 6/52 (12%) of patients in active addiction appeared dependent on only alcohol.

[3] Three recovery stages: early (<1year), sustained (1–5years), and stable (>5years) [39]

[4] Approximately 7/9 (78%) participants in active recovery on OST

OST = Opioid Substitution Therapy

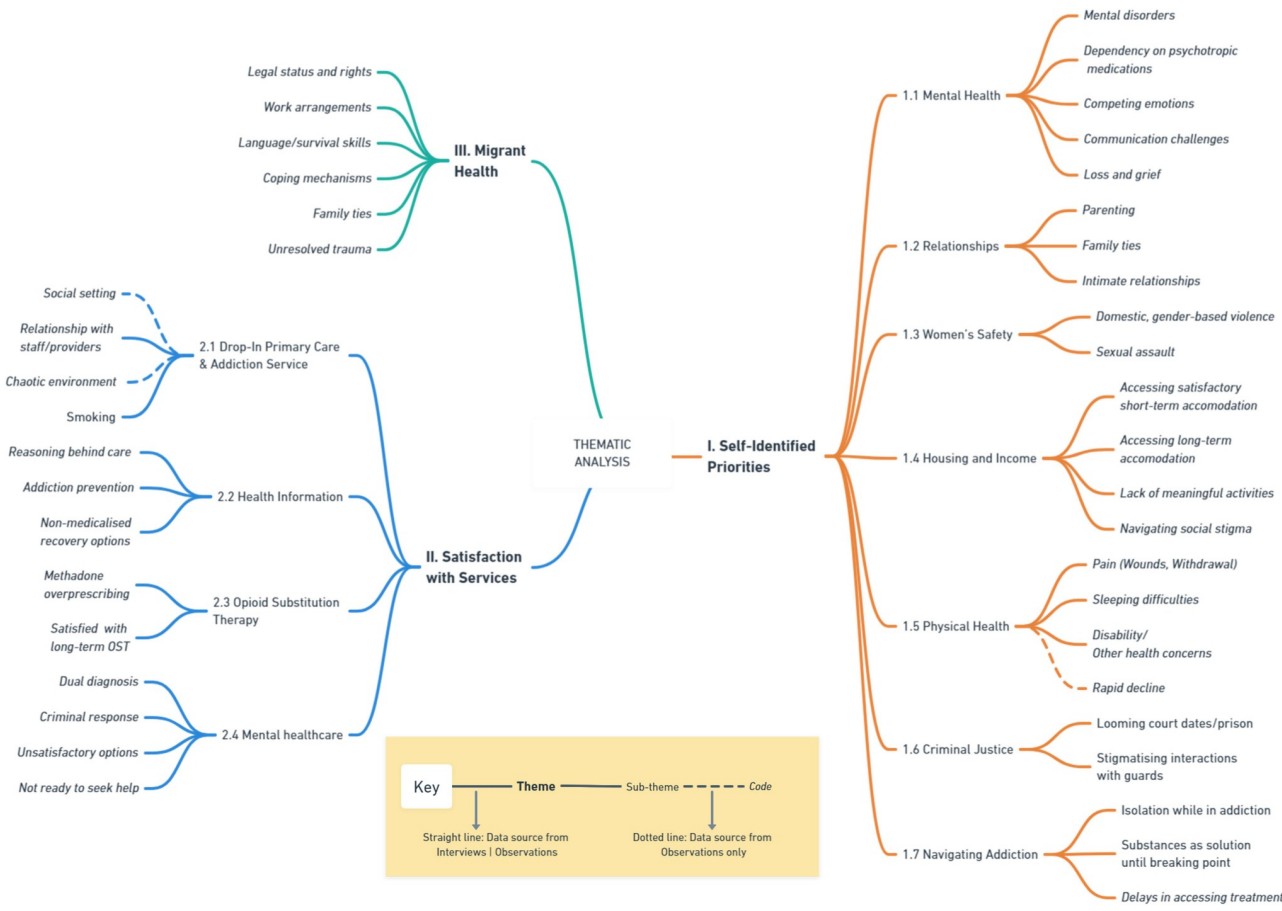

**Fig 1. Emergent themes, sub-themes, and codes.**

participants were sleeping rough (N = 7, 10%) or were in long term accomodation after experiencing homelessness in their lifetime (N = 8, 11%). Of 39 clients with active drug addiction, 32 (82%) were on opioid substitution therapy (OST). Tragically, two participants died of overdose in their hostel during the study observation period. Three clients were waiting to enter a residential treatment programme: 9 clients had completed residential treatment and were in early recovery (<1 year) or sustained recovery (1–5 years) [39] (7 of these continued their OST).

The following sections focus on three main themes emerging from the analysis: self-identified priorities, satisfaction with health services, and migrant health. Fig 1 summarises the broad themes, sub-themes, and codes. Illustrative data extracts from the lead researcher's field notes are provided throughout the text for each sub-theme and labelled by participant pseudonym. Embedded and longer quotations were selected with the aim of invigorating the text and illustrating diverse participants' complex experiences. For conciseness, the use of longer quotations was reserved for findings not often referred to in the literature [38]. The complete coding framework, which provides more granular illustrative examples for each code, is available in S1 Table. To aid comprehension of the results, a glossary of health, legal, and colloquial terms specific to the Irish context is provided in S1 File.

### Theme 1—Self-identified priorities

The broad theme of 'Self-Identified Priorities' refers to priority health concerns identified by participants that tended to fall outside the traditional biomedical model of health [41]. The sub-themes described below emerged as the specific issues most relevant to the health and wellbeing of clinic attendees based on the extent to which they sparked animated discussion with and/or among participants.

**Mental health.**   Clients active in their addiction expressed feeling anxious, depressed, panicky, and dependent upon psychotropic medications for those conditions. Often clients needed 'a smoke' to calm down in moments of distress or anxiety. Clients mentioned feeling isolated due to their addiction and to the severity of traumatic events experienced. Shame, anger, and stress were expressed frequently, at times escalating to suicidal ideation:

> *Eimear says it's hard for her to relate to the women she is around due to the severity of her injuries and experiences. She asks me at one point, "What's wrong with me?" She says she's having nightmares all the time, that she feels panicked and anxious. She says she wants peace, that she's ready to stop trying, that she thinks her kids' lives might be simpler without her.*

Mental health conditions also presented communication challenges. Certain clients remained withdrawn, at times appearing heavily medicated. Others were talkative but difficult to understand due to incoherent ideas. One man had begun hearing voices telling him to do bad things. He didn't want to see anyone about these, reassuring the researcher *"I'll be alright"* (Tristan).

Many clients were grieving the recent loss of a partner, sibling, parent, or close friend, in some cases to suicide. The experience of sharing a room with someone who fatally overdosed in the hostel arose as particularly distressing. One woman explained to a friend, *"I'm the one who found her. I went over to wake her up and saw one side of her face was black. All night I slept with her like that"* (Observed). Compounded losses proved especially overwhelming:

> *Sean suffered from crack cocaine addiction for 30 years but made a complete recovery. In the past year, he turned to alcohol (and eventually alcohol dependence) in relation to his dog being taken away and his sister committing suicide around the same time. He's in the clinic for Librium to help with the anxiousness he has been feeling lately.*

In summary, clients expressed a strong desire to feel better mentally particularly when coping with grief, loss, and proximity to death. Despite being in pain, many were not ready to consider addressing the underlying causes of their mental distress while still navigating the complexities of precarious housing, relying heavily instead on self-medication with street drugs and prescribed psychotropics.

**Relationships.**   Relationships with family and children stood out as essential to participants and evoked strong emotions. Many maintained a positive relationship with their kids and were excited for upcoming outings and time spent together. Others were trying to reduce their substance use so as to reunite with their children. Women who were pregnant or had just had a baby were generally enthusiastic about this, though one mother in recovery expressed feeling overwhelmed by a lack of parenting support: *"If there's one gap in services, it's social workers for single mothers recovering from addiction whose children are returned to them"* (Nora). The difficulties of caring for children with special needs were mentioned (e.g., behavioural issues, autism, attention-deficit/hyperactivity disorder).

Some clients were worried about not being able to get in touch with their families and/or from ties being cut:

> *Ryan is concerned that his family are back to not talking to him again. His nephew's christening is soon but Ryan isn't invited. He says he'll drop off a gift anyway. He says he was using the family's [streaming service] account but that someone changed the password. "Isn't that petty", he says, "The whole family's using it, but they have to change it because of me."*

Phones were frequently broken, stolen, or confiscated, leaving clients upset by having lost contacts. On multiple visits, clients mentioned trying to navigate visiting and caring for elderly or sick relatives. Instances of methadone prescriptions being revised to accommodate these visits were observed by the researcher; something that clients were grateful for each time. Several clients were struggling with recent breakups of long-term relationships. Women noted trying to navigate complex intimate relationships including those with violent partners; with partners whose behaviour had recently changed in relation to escalating drug use; with partners who remained in addiction as they tried to recover; and with partners set to be released from prison: *"A woman tells her friends that 'her man' is getting out next week, saying 'you know what that means.' She feels nervous about this." (Observed)*

Unsurprisingly, when going well, relationships with family, friends, and partners contributed to improved mental health status: when gone awry, they exacerbated mental distress. In the absence of / in addition to strong personal relationships, the researcher observed how supportive relationships with clinic providers and staff that included humour and compassion helped to improve clients' moods.

**Women's safety.** Tying in with relationships was the issue of women's personal safety. Women told the researcher about current threats against their lives by abusive ex-partners:

> *As we're speaking, she starts to break down. She tells me that the man who violently assaulted her just got out of prison on compassionate grounds. "Where is the compassion for me?!" she says. She saw him on the street that morning and had to get off the bus to get sick. (Eimear)*

Clients further detailed how difficult it is to be a woman in homelessness: *"If you have drugs in your bra or panties, those guys aren't afraid to come and find it there" (Cara)*. One woman had been asked to leave a domestic violence shelter when her abuser found out where she was staying. The researcher asked her how she stayed safe after leaving, at which point the woman took out her removable partial denture to reveal a missing tooth. Another client explained that despite being able to stick up for anyone else, she couldn't seem to find the same anger or strength to stick up for herself in abusive situations.

Throughout the study period, female participants drove home what it feels like to have no safe place to go despite being under threat, and the all-consuming place that fear and shame take up mentally in the absence of safety.

**Housing and income.** Some clients were preoccupied with finding a bed for the night, periodically spending long stretches on hold with the public telephone number to call to access emergency accommodation: *"What are they gone for their tea and biscuits?!" (Observed)*. Frustrations ensued when the phone line cut off after an hour and the process had to be repeated. An older client could not access a bed for the night because he was from a different county. He had been sleeping rough for months and was struggling to carry his belongings in a duffel bag while using crutches for a hip injury. Several clients had been waiting years to access more permanent social housing.

Clients able to access beds in hostels were frustrated by the prevalence of drug use within them. One woman waiting for a spot in residential treatment relied on smoking weed to *"at least be doing something" (Ailis)* while her roommates used. For another individual detoxing from alcohol, the lack of private, supportive environment was proving difficult as he went through severe withdrawals. Strict rules (e.g., no paracetamol allowed in hostel) enforced by staff perceived as condescending and inexperienced were of concern. Upon entering homelessness, one man with severe mobility impairments had been assigned a hostel bed on the third floor of a building with no elevator.

Boredom brought on by limited resources and mobility impairment was frustrating to those who found themselves idling all day in homeless accommodation. Some clients were unable to work due to disability or immigration status; others did not want to seek employment for fear of losing disability payments. For clients who were working, satisfactory job choices could be limited due to scheduling conflicts with methadone prescriptions or failing to pass a criminal record check.

Social stigma impeded clients' ability to manage their day-to-day affairs. When interacting with social welfare or other services, several told the researcher about experiences of being hung up on or ignored.

*Sean tells me that he tried to ring social services this week and was hung up on. He shows me a card with a list of numbers to call—at least 10—it's very confusing. He says when he finally got someone on the phone he said, six times, "Please don't hang up on me", but the person did. He tried to set up an account this week at the Credit Union. The girl behind the counter kept asking him questions that didn't make sense for someone in homelessness. He says there's a lot of people in those type of positions that don't know what to make of him.*

In essence, participants described a vicious cycle whereby living conditions in homelessness (e.g., boredom, overexposure to substances) exacerbated substance use which—compounded with high rates of disability and social stigma—impeded future chances of accessing satisfactory employment or housing.

**Physical health.** Despite being in a general practice, only three clients mentioned physical health conditions beyond pain, sleep, or disability. One man proudly informed the researcher that he'd taken his HIV tablets twice daily for over a decade: *"don't forget that undetectable means untransmissible!" (Tyler)*. Another client was trying to get their teeth fixed and a third mentioned dealing with bladder incontinence. Clients often complained of sleep deprivation due to behaviours of roommates in hostels, conditions while sleeping rough, and mental health conditions (e.g., panic, anxiety, post-traumatic stress disorder). The researcher frequently observed clients sleeping in the waiting room.

Physical disability was prevalent, as indicated by frequent use of wheelchairs and crutches, as was mental disability. During in-depth conversations, two women expressed frustration at having lost cognitive function due to injuries perpetrated by former partners: *"I can't focus or communicate things in order anymore" (Eimear)*.

Not infrequently, clients reported to the GP with wounds self-treated with toilet paper, cellotape, plastic shopping bags, or superglue. One man who came in for *"just a knife wound this week, nothing too painful" (Gavin)*, said he would not present to hospital for his injury for fear of having to report the perpetrator and becoming a target when back on the street. Pain was a common theme. Many clients came into the clinic limping or wincing as they sat and stood. Others in varying stages of alcohol detoxifcation experienced tremors of the hands and face as they waited. More vocal clients would call out the extent of their pain if frustrated by wait times. Over the seven month observation period, the researcher observed the rapid decline in

health of several individuals who grew more frail, thin, and withdrawn over time before—in one instance—passing away:

> Tierney is back this week with a key worker. He's not doing well at all. He seems very frail and is sitting on his stretcher, moaning. He has wet himself. The nurse gives him something to drink and makes sure he finishes his breakfast. He says, "I don't feel well" many times. It's very hard to see the difference in him since the last time I chatted with him.

> [Fieldnotes excerpt from 2 months later] I learn from clinic staff that Tierney died last week of overdose.

Despite limited capacity to focus on long-term health and preventive care, as with mental health, clients very much wanted to feel better physically and relied on substances (oftentimes self-prescribed) to cope with physical pain and sleeplessness.

**Criminal justice.** Clients interacted routinely with the criminal justice system, with many having spent time in prison. Legal convictions were of concern when they impeded family responsibilities: *"Keiron says his biggest concern is regarding his upcoming court dates and how he can take care of his kids if he goes to prison."* However, in relation to the difficulties of life outside in homelessness and that *"it's easier to get clean on the inside" (Ryan)*, prison was not always perceived as a negative experience:

> Leonard is set to go into rehab for the first time in January and has over ten court dates lined up in the meantime. If the judges find him guilty, he will go to prison and not to rehab. He doesn't seem too concerned by this; he's been before and says it's not too bad there. He can work out, get free meals and a travel ticket upon release, and all of his friends are there. "I can go see the boys!" he jokes.

Conversely, clients routinely mentioned negative interactions with the guards (Irish police force—An Garda Siochana) involving having belongings confiscated, being insulted, and *"getting battered."* One client described a particularly stigmatising experience at the hands of law enforcement:

> In the past year, Sean was sleeping rough and was awoken by the guards for being "drunk and disorderly", which he doesn't agree that he was, being asleep after all. Sean had a dog for years who was "his baby". When the guards woke him, they took her. They also cut off his pants, which contained his wallet and ID, and never returned these to him. Sean believes someone reported him despite how carefully he looked after his dog, "She was in better shape than me I tell ya, gave her the food off me own plate." He doesn't understand why someone would make false claims against him.

The researcher noticed a tendency for clients to take accountability for their actions—*"The warrants are fair really. I've caused a lot of trouble from the drinking" (Leonard)*—and to expect the same from others. In this sense, the criminal justice system concerned clients most when it was perceived to be unjust. Expected poor treatment from the guards meant that clients generally avoided them: *"A member of staff asks Eimear if she's gone to the guards after running into the man who assaulted her. 'What for?' she says, 'They'll just say I'm some junkie.'"* Consequently, perceived and experienced stigma impeded clients' ability to seek support from law enforcement in moments of crisis.

**Navigating addiction.** Clients explained how their world of homelessness and addiction remains separate from the rest of society:

*I ask Jack if there's anything he wishes more people understood about his life. He thinks for a second, "Oh you know, people walk right past you and don't see you. I counted once. . . 90 people and no one gave a thing. I made a bet with a few of the girls standing around and was right. 8 out of 10 people will ignore you completely; 1 will look you in the eyes and say sincerely that they're sorry but don't have it; and 1 might, <u>might</u> give you something."*

Facing this isolation, that *"people in addiction find companionship in addiction"* (Lorcan) explains in part why for many participants, substances represented a solution rather than concern. Only six clients expressed wanting to reduce the volume of their substance use over the course of the study period. Four of these individuals vocalised wanting help while two others did not seem ready to accept it:

*"I've been in [this city] for years and I hate it. I think it's disgusting. There are drugs everywhere and this addiction I have is a borne on top of me and I can't get rid of it." She says she doesn't see the point of anything anymore but doesn't know how to change or what to do. "There's just no way I'm getting through the next few days sober." (Pamela)*

One man had been working on reducing his methadone dose for months to be able to enter treatment on a Friday in April. When told that he would need to wait until the Monday he was crushed. He told a member of staff, *"if I'm going to use, today's going to be the day" (Observed).* The researcher witnessed firsthand the painful withdrawals and uncertainty associated with resolving to reduce one's substance use. Once clients expressed being ready to seek help, timely access to appropriate addiction care was paramount.

### Theme 2—Satisfaction with health services

The second broad theme encompasses participants' views on and experiences with health services. 'Satisfaction with Health Services' emerged as separate to 'Self-Identified Priorities' because, despite having strong views on particular services, participants rarely viewed their health care as a cause of particular worry or concern. This was an interesting distinction as participants often worried about a particular *condition* (i.e., mental condition, physical pain) but made fewer complaints regarding their access to treatment for that condition nor the systemic inequities which may have brought it on.

**Drop-in primary care and addiction service.**   Over the course of her observations, the researcher observed the extent to which the study clinic was a social environment, with noticable camraderie between staff and clients and amongst groups of acquaintances. Many clients came with a friend or made plans to meet up later. Smoking was a big part of the clinic social scene. Clients frequently shared tobacco and congregated outside to smoke and chat. Word travelled in the waiting room about the quality of various doctors, nurses, addiction services, hospitals, and hostels across the city. The researcher heard one woman dissuade a friend from entering a particular stabilisation facility:*"that place is a kip" (Observed).* Tips and tricks of the trade were frequently shared such as how to access a bed for the night or how to ask for sleeping pills. While clients took comfort in the advice of their peers, the researcher noted several instances of misinformation being spread (e.g., dangers of the Covid-19 vaccine, incorrect requirements to enter residential addiction treatment).

In terms of complaints, the clinic grew more crowded over the researcher's observation period. Some clients were frustrated by wait times and frequently took these frustrations out on staff. Others were frustrated by drug dealing and use in close proximity or by occasional chaotic situations in the waiting room. However the majority of clients, when asked about

their healthcare, reported feeling satisfied with the services they were receiving in the clinic. This mostly came down to the care provided by the medical professionals and operations staff. One client explained that it was the first time she felt that she could be honest with her doctor about her drug use: *"It feels really good to be trusted. It feels good that people here know my name" (Róisín).*

**Information.** Clients noted instances where they lacked desired information regarding their healthcare. One man—unaware that Librium was not meant to be taken with alcohol—said that he had taken 9 tablets, drunk a bottle of vodka, and wound up in an altercation with the guards. Another man didn't understand why he'd been asked to return to get his bloods done multiple times. Another still was completing a benzodiazepine detox and unsure of sure why he was being handed stacks of pills each visit. He complained to a friend, *"I'm trying to get off this stuff, I don't want to be on these forever." (Observed)*

Clients desired more proactive outreach for those in addiction. One man wished that more of an effort was made to go into communities and target individuals in early stages of substance use: *"Make sure to tell the doctor that. You have to get to the young people. It's really important" (Timmy).* Another hoped to see more information and outreach regarding non-medicalised options for recovery:

> *Lorcan says the pain of facing your trauma is not as bad as the pain of addiction. He wishes people knew this. When GPs hear somebody say 'I'm fed up. I can't do this anymore', he wishes they would point them towards stabilisation, residential treatment, 12-step. He says that a lot of people, even GPs, don't know about the services available. They don't know that there is a way to face the pain besides drugs and alcohol. In addiction, those seem like the only option. He wants people to know that it's not.*

Then again, during peer debriefing clinic partners explained their hesitancy to promote residential addiction treatment services that are in short supply and create risk of relapse and overdose upon discharge into homelessness.

**Opioid substitution therapy.** Despite over half of the study population mentioning or being observed picking up their methadone / suboxone prescriptions during the study period, OST was a contentious topic amongst participants:

> *Jack says that if he could change one thing, it would be how hard it is to get off methadone. He explains that now, a person starts on a lower dose until, say a few weeks in, they use street drugs. The doctors then raise the methadone dose. To him, this doesn't make sense. If a person was having withdrawals so severe that they needed to take illicit drugs in addition to their methadone, that would have happened early on, not several weeks in. He doesn't think raising the dose is the solution because it's very hard to get off.*

Two clients said that when they expressed a need for help with their addiction, their private GP had only wanted to talk about methadone without mentioning other options. One found a new GP who was willing to follow that client's own plan for recovery. Conversely, two women who had been on methadone for four months and nine years respectively mentioned that it was going well. Across her visits, the researcher observed clear improvements in stability in those on lower doses of methadone compared to those just starting.

The topic of convenience came up. For one client, picking up methadone prescriptions clashed with working hours: *"I've a key worker that can pick it up for me but she's not always around" (Timmy).* Another individual missed her methadone appointment on a Wednesday, ran out Sunday, and resorted to street drugs on Monday. Two women on suboxone liked its

convenience—*"I can carry it around in my purse" (Aileen)*—but one said she could have used extra support when transitioning from methadone to suboxone due to the severity of withdrawals. On one clinic visit, the researcher overheard clients discussing how they were being prescribed methadone for pain but didn't want it: *"I've never taken a drug in me life, I don't want that stuff" (Observed).*

**Mental healthcare.** Clients described negative experiences with mental healthcare including ongoing issues with dual diagnosis within a health system that treats addiction as separate to health:

> *Jack says that he went to a psychiatrist a day that he was suicidal and was sent back to the streets with two pills, one for the night and one for the morning. He took both immediately. The psychiatrist said that the suicidal thoughts were linked to his drug use and not his mental health. "That doesn't make sense because I've been using drugs since I was a teenager but I'm only suicidal since my partner kicked me out."*

Another woman faced a similar dilemma, *"I can't get psych medication until my addiction is sorted but the more I get my addiction sorted, the more it's bringing up old emotional stuff that I need help dealing with" (Alannah).* In other instances, mental health was responded to as a law enforcement issue as demonstrated by this exchange between a client and the researcher:

> *She tells me of one episode of psychosis so bad that she ended up covered in blood running in and out of cars. The police came but didn't call the ambulance. "Why?" I ask. She laughs, "You think they'd come for me?! Having a psychosis, supposedly on drugs? That could take ages." Instead, she was put into a jail cell until she was feeling well enough to leave. The police know her. "They did charge me," she says, "obstruction of traffic." (Nora)*

Oftentimes, desired treatment options such as counselling specific to loss, death, and grief were not available:

> *Teresa says her rehab facility charged [nearly €100] a week and didn't provide grief counselling, or any counselling. "I **know** grief", she says. She is very upset about the loss of her brother. "Broke me heart, broken as it is."*

In parallel, the researcher observed the importance of timely and convenient care. One woman had been seeing a therapist based in her hostel but stopped going when she changed housing. Another expressed a strong desire to speak to the clinic's drop-in psychotherapist but never returned to avail of the service on the day it was available. Others still had internalised the belief that they were beyond help. When describing her past experience seeing a domestic violence counsellor, participant Eimear explained *"I needed more help than she could give me."* Participant Alannah expressed her own hesitancy towards therapy: *"Oh you know me, once you got me talking I just wouldn't stop."* There were positive stories too. A woman who had been assaulted got a letter from the GP to see a psychiatrist and felt better being able to talk to someone.

### Theme 3—Migrant health

The final broad theme 'Migrant Health' emerged as separate to the others as migrant participants dealt with a set of competing priorities unrelated to substance misuse. The researcher spoke with four African individuals attending the clinic who had obtained European citizenship in other countries but—due to precarious tenancy agreements and lack of affordable

housing—were living in temporary homeless accommodation in Ireland. For these individuals, primary concerns related to working long hours, navigating long commutes to work, and applying for the Housing Assistance Payment (HAP). Other, asylum seekers faced unique challenges relating to traumatic, months-long journeys; lack of beds in Direct Provision and having to sleep rough upon arrival in Ireland; not being allowed to work before the processing of the asylum application and having to survive on a meagre weekly allowance; and fearing for the safety of family back home.

> *Ken has been in Ireland a month or two. He literally fled for his life. His wife and children, his family, are still [in his home country]. It took him months to get to Ireland. He says it was the hardest period of his life. When he arrived in Ireland, he registered with International Protection Office only to find out that he wouldn't receive housing until his application goes through, which can take months. He is homeless. The men with him are homeless. I ask where they manage to sleep. It seems like a mix between hostels, sleeping rough, or trying to crash with friends who have housing. Ken is not allowed to work as an International Protection Applicant and yet he's only receiving €38.80 a week to live on. This is to find housing, food, transport, everything.*

These challenges impacted health. One man, despite being diagnosed as pre-diabetic in his home country, had resorted to eating lots of candy since arriving in Ireland as a small comfort.

All migrant participants were trying to navigate living far from family and legally bringing their spouses and children to Ireland. Language barriers were present. On multiple occasions, groups of migrants attended the clinic with a 'spokesperson' (usually a friend) to serve as an interpreter. While many of the Irish clients seemed familiar with one another, groups of clients from other countries typically remained to themselves and spoke in their respective languages. Unprompted, a man experiencing homelessness and addiction who had grown up in Ireland described how refugees may lack the skills necessary for surviving on the streets:

> *Gavin said the refugee recognised him on the street and asked for help finding some cardboard and a marker to write up a sign saying that [the refugee] was from Ukraine and homeless. Gavin found the supplies and started helping him ask people for money. He says the man was shocked when they made about €10 in five minutes. He explained that refugees coming in don't have the street skills he's had to use all his life.*

Many of migrant clients' priority concerns intersected with those of the Irish cohort; they battled stress, stigma, distance from family, painful physical health conditions, and barriers to accessing satisfactory housing. Concerns diverged, however, in that migrant clients less frequently expressed feeling mentally unwell despite going through traumatic experiences, were battling language barriers, and that—for asylum seekers—survival in Ireland was made especially difficult by a lack of right to work or receive social welfare.

## Discussion

### Proposed conceptual model

The aim of this study was to explore the perspectives of PEH attending an urban primary care and addiction service on their priority health and healthcare needs. To best encapsulate these perspectives, an inductive approach grounded in social constructivism was used to build a model of participants' self-expressed needs, and barriers / facilitators to having them met, based on the themes emerging from the data (Fig 2). The pathway towards health and

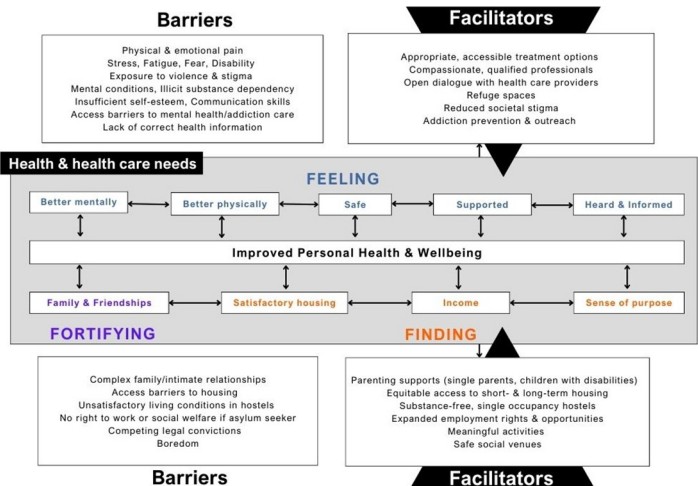

**Fig 2. Social constructivist conceptual model of the priority health and health care needs of primary care patients experiencing homelessness.**

wellbeing depicted is based entirely on participants' concerns and factors they consider important; tobacco consumption, for example, is not included as clients considered this to be a comfort not concern.

Thematic analysis pointed to a social constructivist model of intersecting health and healthcare needs relating to Feeling, Finding, and Fortifying ('3F'). Clinic attendees want to feel better mentally, to feel better physically, and to feel safe, supported, heard, and informed. The barriers expressed to feeling these things have been widely reported in the literature. Physical and emotional pain [42–45], competing emotions [44], mental health conditions [43–50], violence [44, 45, 47, 50], stigma [44, 47, 51, 52], and disability[43, 50, 53] are pervasive both prior to, and during experiences of homelessness, compounded with inequitable access to satisfactory health information and care [47, 54]. Unique to this study, participants proposed their own solutions for helping to overcome these challenges including appropriate and accessible mental health and addiction care that involves grief counselling, psychotropic medications, recognition of dual diagnosis, and faster access to residential addiction treatment; receiving compassion from qualified health care professionals and staff; holding transparent conversations with health care providers about treatment goals and options; increased availability of refuge spaces; wider societal understanding of addiction and homelessness; and increased focus on addiction prevention and outreach for those in early stages of substance use or those wishing to reduce their usage.

In terms of the lower two 'Fs' in the model, in light of complex relationships and legal situations [44, 45, 47, 49, 51, 52, 55–57] coupled with housing and other socioeconomic inequities [43–46, 52, 53, 55, 58], clinic attendees seek to find satisfactory housing, income, and a sense of purpose, and to fortify their families and friendships. Self-expressed facilitators of achieving these goals include increased parenting supports, particularly for single parents in recovery and parents of children with disability; safe, social venues and participation in meaningful activities; expanded employment rights and opportunities, particularly for asylum seekers; equitable access to temporary and permanent housing for individuals with disability and / or from different geographic jurisdictions; and substance free, single occupancy hostel rooms.

The model depicts how these facilitators contribute to the attainment of the '3Fs' and how meeting one health need has a knock-on effect on each of the others. Fortifying family and

friendships, for example, contributes to finding a sense of purpose. Finding a sense of purpose contributes to feeling better mentally which, in turn, can build confidence to seek education or employment and find income. These intersecting elements contribute to meeting the over-arching need of improved personal health and wellbeing which, by strengthening capacity to overcome barriers and avail of existing supports, perpetuates a virtuous cycle towards improved health.

## Implications for policy

Moving forward, clients' views underline a need to address the ongoing societal stigma towards homelessness and addiction that trickles down to impede fair treatment within health and social services, and the criminal justice system. Participants' experiences of trying to access long-term housing demonstrated that systemic barriers to housing persist and must be addressed including difficulty navigating the system for those with physical and mental health conditions. For as long as permanent housing remains inaccessible, the number of single or family rooms in temporary accommodation that provide privacy and respite from exposure to illicit substances should be expanded and access barriers to these rooms reduced by implementing a more efficient phone system (e.g., more operators, a callback alternative to waiting on hold), ensuring that assigned rooms accommodate mobility impairments, and removing the requirement to show proof of residency in order to access a bed within a certain jurisdiction. These improvements are especially important as the anecdotes participants shared regarding the inaccessibility and poor quality of temporary housing drove home the impracticality of expecting individuals to focus on improving any aspect of their health in the absence of safe and stable accommodation.

In terms of mental health and addiction services, participants expressed a need for counselling specific to loss, death, and grief. One recommended response would be to create space and opportunity for memorial services within housing services or public settings to reduce social exclusion and disenfranchised grief, and provide a form of advocacy [59]. In Seattle, homeless Housing First residents recommend integrating grief counselling into single-site or shelter settings where residents who live in close proximity and share communal spaces may be deeply affected by the death of a fellow resident [60]. Requirements that PEH reduce their substance use in order to access mental healthcare should be reconsidered in light of over-whelming evidence that this is detrimental to health [61]. With respect to clients' desire for upscaled addiction prevention efforts, research has shown that community-based programs that deliver a coordinated, comprehensive message about prevention can be effective in preventing adolescent substance use [62]. Local drug and alcohol task forces are charged with implementing this type of drug prevention intervention in Ireland [63]; that clients taking part in this study appeared to have fallen through the cracks of these programmes points to a need to review their effectiveness and reach. On the treatment end of the spectrum, clients' experiences are in line with existing evidence that there are not enough residential addiction treatment beds nationally to ensure timely care [11].

Regarding social services, there is an urgent need for more refuge spaces with a particular regard for women for whom experiences of domestic, sexual, and gender-based violence are both a cause and consequence of chronic homelessness. The finding that social care tapers away prematurely once parents enter recovery and find more stable housing has not been widely reported and may indicate a need to review eligibility requirements for social work services. In terms of more casual social supports, walk-in health services often serve as social hubs for PEH who attend clinics with peers, look out for each other, and encourage others to seek help to feel better [51]. Clients desire additional safe, social venues and activities that provide

meaningful ways to combat boredom and strengthen bonds formed while in homelessness. Successful examples from the literature include opportunities to come together for education, volunteering, recreation, and work [52]. That applicants for international protection currently have to wait 6 months before becoming eligible to work in Ireland has serious social, financial, and in turn health, implications [64]. Current policy mandates a weekly allowance of €113.80 for asylum seekers if there are no spaces in International Protection Accommodation Services (IPAS) [65] but clients taking part in this study were receiving significantly less despite being unaccommodated. As such, the distribution and suitability of current allowances which fall well below the Irish living wage [66] should be reviewed in the absence of equal employment opportunities.

## Implications for practice and research

That women in homelessness present to community health clinics after experiences of sexual, domestic, or gender-based violence creates opportunity in crisis to link in with refuge spaces and psychological supports provided they are available. Recognising that that asylum seekers and other migrants often lack information about available supports or find them difficult to access and navigate [47, 67], health clinic staff may also help by directing recently arrived, non-Irish clients to targeted non-governmental organisations. Clients' most prevalent recommendation for practice was that more open dialogues be established with health care providers. Yet, GPs expressed trying to balance parallel responsibilities to maintain non-maleficence from prematurely promoting recovery while also respecting the autonomy of PEH to learn about and choose treatment options in line with their goals. To navigate this dichotomy, providers may start by mentioning avenues for clients to participate in meaningful social activities and available community-based treatments [68] with a view that regardless of substance use outcomes, social support and perceived choice are good for health [57, 69]. Explaining in part the strong and differing opinions clients expressed on methadone, research shows that service users with opioid use disorder generally have limited understanding of the chronicity of their illness, the likelihood for long-term OST or the rationale for using methadone as a medical intervention in their disease management [70]. In line with clients' desire to be heard and informed, GPs might consider having regular conversations with patients on OST to create realistic expectations, discuss the rationale behind the treatment, and ascertain if any changes in medication are desired that might be safely made.

Establishing open dialogue with all clients will require a better understanding of effective strategies for overcoming communication challenges due to language barriers, mental conditions, side effects of antipsychotics, and/or impaired cognitive function. Participants affected by the latter condition expressed frustration during this study at the memory loss and inability to focus experienced after head injury. Accordingly, exploring the feasibility and acceptability of adapting rehabilitation interventions for traumatic brain injury (TBI) for those in homelessness could be of value [71]. Finally, questions arose regarding clients' views on appropriate mental health care. For example, despite high rates of traumatic experiences, migrant clients less frequently expressed concern over their mental health, begging the question of how psychosocial supports for those in homelessness might be viewed / prioritised differently across different cultures. As a whole, the unique health needs of individuals experiencing homelessness after migrating from diverse countries and conflicts are not well understood and demand further research.

### Limitations

This study was located in one clinic providing accessible, walk-in primary care and addiction services which impacts findings in several ways. First, findings capture the views of a population linked with health services. While crossover was identified with views expressed by homeless service users in Canada, the US, the UK, Sweden, Norway, Australia, New Zealand, and South Africa [42–46, 48–50, 51–56, 58], these findings may not be transferrable to individuals who are not accessing care nor to locations where access to primary care is more limited for those in homelessness. Second, in some instances clients may have been hesitant to share views on the care they were receiving in the same clinic where interviews were taking place. Because participants who participated in in-depth conversations were referred to the research team by clinic staff with whom they had an established rapport, it is likely that the sample captures more favourable views of the primary care and addiction service than may have otherwise been the case. While not recording the interviews allowed researchers to capture organic and informal conversations, certain nuances of clients' perspectives may have been missed in the writing up of field notes. As well, some clients were hard to understand due to mumbling, slurring, or incoherent ideas and not all of what they shared could be documented. This served as a poignant reminder to the researchers that often, individuals who need support the most are less likely to be able to communicate their needs and/or advocate for themselves.

## Conclusions

Using an ethnographic approach, this study sought to explore the views of PEH attending a walk-in primary care and addiction service on their own health and healthcare needs. Many of the priority concerns identified by clients experiencing homelessness are not dissimilar from those of the general population. Clients need to feel better mentally and physically, to feel safe, to fortify bonds with family and friends, and to find a sense of purpose. The difference is in the gravity of these needs observed both prior to, and during experiences of homelessness exacerbated by disproportionate exposure to loss, fear, pain, disability, fatigue, stigma, and systemic barriers to accessing safe and stable housing.

An encouraging finding is that clients taking part in this research are satisfied with their ability to access primary care and harm reduction services in a social environment where positive exchanges with friends and providers take place. Conversely, barriers to accessing mental health and addiction services persist, including the internalised belief that one is beyond help, lack of access to information on available services and their entry requirements, and lingering stigma within a health system that treats addiction as separate to health. Recommendations for practice include holding more regular and open conversations with clients experiencing homelessness during primary care consultations about the care they are receiving, its rationale, and whether or not changes are desired that can be safely made. In consultation with service users and providers, policymakers should prioritise expanding access to refuge spaces, parenting supports, and residential stabilisation, treatment, step-down, and aftercare beds for PEH. The health needs of migrants and asylum seekers entering homelessness in Ireland are urgent and, as they are not yet widely understood, should be prioritised in future research.

## Supporting information

**S1 Dataset. Fieldnotes data excerpts and participant characteristics.**
(XLSX)

**S1 Table. Thematic analysis codebook with illustrative quotes.**
(DOCX)

**S1 File. Glossary of Irish health, legal, and colloquial terms.**
(DOCX)

## Acknowledgments

We would sincerely like to thank community health clinic partners for contributing their practical and clinical expertise to this project and facilitating the project space and participant recruitment. We acknowledge that this project was made possible in large part thanks to the trust built over time between clinic staff and clients which allowed us to be accepted more quickly into the space as research colleagues.

## Author Contributions

**Conceptualization:** Carolyn Ingram, Conor Buggy, Carla Perrotta.

**Data curation:** Carolyn Ingram.

**Formal analysis:** Carolyn Ingram, Isobel MacNamara.

**Investigation:** Carolyn Ingram.

**Methodology:** Carolyn Ingram, Conor Buggy, Carla Perrotta.

**Supervision:** Conor Buggy, Carla Perrotta.

**Visualization:** Carolyn Ingram.

**Writing – original draft:** Carolyn Ingram.

**Writing – review & editing:** Carolyn Ingram, Conor Buggy, Isobel MacNamara, Carla Perrotta.

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
