## [Decision Letter · Decision Letter 0]

22 Mar 2024

PONE-D-24-06202“Just a knife wound this week, nothing too painful”: an ethnographic exploration of how homeless clients attending an urban primary care and addiction service view their own health and healthcarePLOS ONE

Dear Dr. Ingram,

Thank you for submitting your manuscript to PLOS ONE. After careful consideration, we feel that it has merit but does not fully meet PLOS ONE’s publication criteria as it currently stands. Therefore, we invite you to submit a revised version of the manuscript that addresses the points raised during the review process.

We look forward to receiving your revised manuscript.

Kind regards,

Prof. Anat Gesser-Edelsburg, Ph.D.

Academic Editor

PLOS ONE

Journal Requirements:

Reviewers' comments:

Reviewer's Responses to Questions

**Comments to the Author**

1. Is the manuscript technically sound, and do the data support the conclusions?

Reviewer #1: Partly

Reviewer #2: Yes

2. Has the statistical analysis been performed appropriately and rigorously? 

Reviewer #1: N/A

Reviewer #2: N/A

3. Have the authors made all data underlying the findings in their manuscript fully available?

Reviewer #1: No

Reviewer #2: Yes

4. Is the manuscript presented in an intelligible fashion and written in standard English?

Reviewer #1: Yes

Reviewer #2: Yes

5. Review Comments to the Author

Reviewer #1: It is a good job but it requires a lot of work to be able to accept it and publish it.

There are important limitations in data collection that are not addressed.

Observation, what was observed, how the information was systematized?

Due to the procedure in which the conversation was carried out, it does not differ much from the processes for conducting an interview (audiotaped) except for the issue of recording. What difference does it really make between recording or not? This poses many limitations in the quality of the data, how was it resolved? Note taking can be as or even more intrusive than recording. Memory bias? How was it resolved?

What strategies were followed to guarantee the quality of the data collected?

There were no references from the informants to other possible informants (snow ball sampling). Was there any other strategy to identify informants? Did you only trust the intuition and decision of the clinic staff? How did this decision influence the type of informants and the quality of the data?

Taking notes regarding what a person says is already an exercise in synthesis and reflection that implies analysis. How was this issue worked on and resolved in the analysis carried out by two researchers? Was there any discussion about decision making regarding note taking?

The presentation of the analytical process in the form of bullets seems to oversimplify it. I suggest a rich description of the process and the way decisions were made.

Was the same data treatment given to the conversations as to the observation? Were the codes the same for conversations as for observation? How was this achieved when it is difficult to account for experiences through observation? How did the observation allow us to know the housing status, living with relatives and other characteristics? Are they the same interview participants?

How is figure 1 constructed? The lines have no direction. Analysis of the figure is missing in the text.

The identification of needs lacks a qualitative analysis. Was it quantified according to the number of times it is mentioned? How those needs were defined. How is a count of mentions justified according to the analytical framework proposed by the study?

Much of the analysis in the following sections is supported by the number of informants who mention the topic. They even refer to the number of people who say it or refer to the majority. Is the relevance of the topics only due to frequency or how was the decision to include them made?

I do not find sufficient elements to assess the relevance of the topics beyond the number of people who mention them. At least it is not present in the document. In that sense, how did you choose the interview excerpts that are presented?

There is an important difference in the development and inclusion of testimonies between topics and in general the analysis remains at a descriptive level. There is no closure in the different sections of the results.

Where do the subtopics come from? How do you get to them when only three were presented as relevant without mentioning the important list of their topics?

This work lacks the depth sought through an ethnographic study. It lacks a conceptual framework that allows relating the findings of the study no matter how much of an ethnographic approach it has sought to have. The fact of posing a question and a problem suggests having a frame of reference. Nor are there elements in the analysis that suggest or allow us to identify the framework proposed by social constructionism.

It would be very interesting to know why they refer to informants as “clients.”

The discussion is little supported by the results presented given the superficiality of the analysis. The same with the conclusions and recommendations they make.

Reviewer #2: I would like to thank the authors for giving me the opportunity to review this high-quality work.

The article is interesting and takes a qualitative approach to a subject relating to health services and addiction. The design is particularly well suited to exploring the user's point of view. I recommend this article for publication in Plos One with some minor comments.

I have a few comments to make to improve the work. I think that the results in terms of health policy implications could be highlighted more in the discussion. I think that if these results are interesting, how can they be articulated with the organization of care from a practical point of view? How can these results enlighten health policy regulators?

6. PLOS authors have the option to publish the peer review history of their article (what does this mean?). If published, this will include your full peer review and any attached files.

Reviewer #1: No

Reviewer #2: No

---

## [Author Response · Author response to Decision Letter 0]

25 Apr 2024

Please see the Response to Reviewers document in the Attached Files.

---

## [Decision Letter · Decision Letter 1]

30 Apr 2024

PONE-D-24-06202R1“Just a knife wound this week, nothing too painful”: an ethnographic exploration of how homeless clients attending an urban primary care and addiction service view their own health and healthcarePLOS ONE

Dear Dr. Ingram,

Thank you for submitting your manuscript to PLOS ONE. After careful consideration, we feel that it has merit but does not fully meet PLOS ONE’s publication criteria as it currently stands. Therefore, we invite you to submit a revised version of the manuscript that addresses the points raised during the review process.

We look forward to receiving your revised manuscript.

Kind regards,

Prof. Anat Gesser-Edelsburg, Ph.D.

Academic Editor

PLOS ONE

Reviewers' comments:

Reviewer's Responses to Questions

**Comments to the Author**

1. If the authors have adequately addressed your comments raised in a previous round of review and you feel that this manuscript is now acceptable for publication, you may indicate that here to bypass the “Comments to the Author” section, enter your conflict of interest statement in the “Confidential to Editor” section, and submit your "Accept" recommendation.

Reviewer #1: All comments have been addressed

Reviewer #2: All comments have been addressed

2. Is the manuscript technically sound, and do the data support the conclusions?

Reviewer #1: Partly

Reviewer #2: Yes

3. Has the statistical analysis been performed appropriately and rigorously? 

Reviewer #1: N/A

Reviewer #2: N/A

4. Have the authors made all data underlying the findings in their manuscript fully available?

Reviewer #1: No

Reviewer #2: Yes

5. Is the manuscript presented in an intelligible fashion and written in standard English?

Reviewer #1: Yes

Reviewer #2: Yes

6. Review Comments to the Author

Reviewer #1: I appreciate the opportunity to review this work again and I see that there is good work by the authors to respond to the observations and suggestions. However, there are still very little developed subtopics (results) such as the case of “criminal justice”; “navigating addiction”. I consider that if there is no possibility of developing them in depth and they really have relevance and weight in the results as well as in the analysis, then leave them out. It is not a simple decision, but I suggest the authors to reflect on the necessity to present stronger evidence.

The biggest detail I see in this version is that social stigma is mentioned in a superficial way and the evidence presented is not compelling to account for such a complex phenomenon that requires exploring in depth. There are not enough theoretical elements in the analysis to support the relevance of the issue of stigma, especially in these especially vulnerable groups. In this way, the proposals made for both research and public policies are not supported by the results. This is quite delicate, especially if we consider the great weight that stigma has in the response to needs and how it operates as an important barrier in accessing services and care.

For this reason, in my previous review I commented that I find the analysis somewhat superficial, and this is explained by the large number of topics that are to be presented and that were explored, limiting the deepening of the understanding of the central themes. Without a doubt, the fact of not having audio recorded the interviews also limits the quality of the data since the interviewer distracts his attention by taking notes and recording verbatim the speech of the informants. I think there is also greater elaboration in this sense not only in the limitations but in the section that was added on quality based on Guba-Lincoln's elaborations.

Overall the paper is much better and if the authors can address this issues it can be very publishable work.

Reviewer #2: (No Response)

7. PLOS authors have the option to publish the peer review history of their article (what does this mean?). If published, this will include your full peer review and any attached files.

Reviewer #1: No

Reviewer #2: No

---

## [Author Response · Author response to Decision Letter 1]

21 May 2024

Please see the attached 'Response to Reviewers' document.

---

## [Decision Letter · Decision Letter 2]

3 Jun 2024

PONE-D-24-06202R2“Just a knife wound this week, nothing too painful”: an ethnographic exploration of how homeless clients attending an urban primary care and addiction service view their own health and healthcarePLOS ONE

Dear Dr. Ingram,

Thank you for submitting your manuscript to PLOS ONE. After careful consideration, we feel that it has merit but does not fully meet PLOS ONE’s publication criteria as it currently stands. Therefore, we invite you to submit a revised version of the manuscript that addresses the points raised during the review process.

We look forward to receiving your revised manuscript.

Kind regards,

Anat Gesser-Edelsburg, Ph.D.

Academic Editor

PLOS ONE

Journal Requirements:

Reviewers' comments:

Reviewer's Responses to Questions

**Comments to the Author**

1. If the authors have adequately addressed your comments raised in a previous round of review and you feel that this manuscript is now acceptable for publication, you may indicate that here to bypass the “Comments to the Author” section, enter your conflict of interest statement in the “Confidential to Editor” section, and submit your "Accept" recommendation.

Reviewer #3: (No Response)

2. Is the manuscript technically sound, and do the data support the conclusions?

Reviewer #3: Yes

3. Has the statistical analysis been performed appropriately and rigorously? 

Reviewer #3: N/A

4. Have the authors made all data underlying the findings in their manuscript fully available?

Reviewer #3: Yes

5. Is the manuscript presented in an intelligible fashion and written in standard English?

Reviewer #3: Yes

6. Review Comments to the Author

Reviewer #3: Title – ‘homeless clients’ can be perceived as stigmatising and there is a push to use non-stigmatising, person-centred language like ‘people experiencing homelessness’ which should be considered in the title and throughout the paper.

Abstract

• ‘people in homelessness’ seems a bit of a strange term – people experiencing homelessness is more common.

• Usually an abstract would include some background/context to provide readers with information about why the study was needed/conducted, so I suggest a few sentences are added here to provide this detail. This would then contextualise the final sentence to explain why their needs should be prioritised.

Introduction

• You use the term ‘people experiencing homelessness (PEH)’ here so this should be used throughout including in title and abstract.

• Bracket missing on page 9, line 43.

• It would be good to mention the health needs assessment methodology in the abstract too.

Methods

• Under participants and setting it would be helpful to have a bit more detail regarding the clinic in which the study was conducted – where located, how funded etc (especially for international readers unfamiliar with the Irish healthcare system).

• Purposive, critical case sampling – it would be good to have more detail, particularly in terms of what criteria were used in terms of purposive sampling and how this was achieved.

• Was the reflective notes template based on others’ work or created for this study? It would be good to make this clear and add any references, if relevant.

• It would be good to discuss if informed consent was granted for the casual conversations and if not, why not. The same would be the case for observations.

Findings

• A note re. language on page 19, line 346 – alcoholism is an outdated and stigmatising term and should be avoided. I’d encourage you to change to ‘problem alcohol use’ or ‘alcohol dependence’.

• Page 21, line 417 – suggest adding a definition to ‘garda vetting’ for non-Irish readers.

• There are also several places where language used is not familiar with international readers (battered, kip) so a glossary might be useful.

• The findings are really interesting and highlight the challenges of participants’ lives.

• I would suggest the authors rethink the inclusion of theme 3 – it is somewhat different to the rest of the findings and is far less detailed – themes 1 and 2 are 7 and 4 pages each, respectively and theme 3 is less than 2 pages. It feels very different to the rest of the findings and doesn’t really add as much depth.

Discussion

• Very detailed and comprehensive discussion.

7. PLOS authors have the option to publish the peer review history of their article (what does this mean?). If published, this will include your full peer review and any attached files.

Reviewer #3: **Yes: **Dr Hannah Carver

---

## [Author Response · Author response to Decision Letter 2]

17 Jun 2024

Thanks very much for your timely and helpful feedback. We've responded to each point made in the attached 'Response to Reviewers' file.

---

## [Editor Report · Decision Letter 3]

25 Jun 2024

“Just a knife wound this week, nothing too painful”: an ethnographic exploration of how primary care patients experiencing homelessness view their own health and healthcare

PONE-D-24-06202R3

Dear Dr. Ingram,

We’re pleased to inform you that your manuscript has been judged scientifically suitable for publication and will be formally accepted for publication once it meets all outstanding technical requirements.

Kind regards,

Prof. Anat Gesser-Edelsburg, Ph.D.

Academic Editor

PLOS ONE
---

## [Editor Report · Acceptance letter]

28 Jun 2024

PONE-D-24-06202R3 

PLOS ONE

Dear Dr. Ingram, 

I'm pleased to inform you that your manuscript has been deemed suitable for publication in PLOS ONE. Congratulations! Your manuscript is now being handed over to our production team.

Kind regards, 

on behalf of

Prof. Anat Gesser-Edelsburg 

Academic Editor

PLOS ONE